# Hydroxytyrosol Inhibits Protein Oligomerization and Amyloid Aggregation in Human Insulin

**DOI:** 10.3390/ijms21134636

**Published:** 2020-06-30

**Authors:** Ivana Sirangelo, Margherita Borriello, Silvia Vilasi, Clara Iannuzzi

**Affiliations:** 1Department of Precision Medicine, Università degli Studi della Campania “Luigi Vanvitelli”, Via L. De Crecchio 7, 80138 Naples, Italy; ivana.sirangelo@unicampania.it (I.S.); margherita.borriello@unicampania.it (M.B.); 2Institute of Biophysics, National Research Council, Via Ugo La Malfa, 153, 90146 Palermo, Italy; silvia.vilasi@cnr.it

**Keywords:** amyloid aggregation, human insulin, hydroxytyrosol, amyloid toxicity

## Abstract

Hydroxytyrosol (HT), one of the main phenolic components of olive oil, has attracted considerable interest for its biological properties, including a remarkable antioxidant and anti-inflammatory power and, recently, for its ability to interfere with the amyloid aggregation underlying several human diseases. We report here a broad biophysical approach and cell biology techniques that allowed us to characterize the molecular mechanisms by which HT affects insulin amyloid aggregation and the related cytotoxicity. Our data show that HT is able to fully inhibit insulin amyloid aggregation and this property seems to be ascribed to the stabilization of the insulin monomeric state. Moreover, HT completely reverses the toxic effect produced by amyloid insulin aggregates in neuroblastoma cell lines by fully inhibiting the production of toxic amyloid species. These findings suggest that the beneficial effects of olive oil polyphenols, including HT, may arise from multifunctional activities and suggest possible a application of this natural compound in the prevention or treatment of amyloid-associated diseases.

## 1. Introduction

Amyloid aggregates are associated with several pathological conditions including neurodegenerative diseases, such as Alzheimer’s and Parkinson’s disease, infectious prion diseases, non-neuropathic systemic amyloidosis, and even type II diabetes [1]. Therefore, the identification of new compounds able to interfere with amyloid aggregation and the related cytotoxicity is considered an advantageous approach for developing novel therapeutic strategies. In this respect, plant polyphenols have attracted considerable interest as they affect different steps in the pathogenesis of amyloid diseases—the aggregation pathway of the involved protein/peptide, the inflammatory response, the alteration of the proteostatic network, and/or the oxidative stress. These molecules also share key chemical features that explain why most of them, although chemically different, induce similar effects [2,3,4]. Also, it has been reported that the Mediterranean diet, naturally enriched in plant polyphenols, is effective against age-related deterioration and improves aging-associated degenerative diseases and neurological deficits [4,5,6].

Extra-virgin olive oil, a key component of the Mediterranean diet, contains high levels of phenolic compounds, including oleuropein aglycone (OleA) and its main metabolite, hydroxytyrosol (3,4-dihydroxyphenylethanol, HT) that have attracted considerable interest for their antioxidant and anti-inflammatory power [6,7,8,9]. HT (Figure 1) is also highly concentrated in red wine and can be endogenously produced via dopamine metabolism [10,11]. Recently, OleA and HT have been shown to affect the amyloid aggregation process in different model proteins in vitro although the molecular mechanisms have been poorly characterized [12,13,14,15,16,17,18,19,20,21]. Because only little information is available for HT compared to OleA and considering that HT is one of the main metabolites of OleA, further studies could be useful for a better understanding of the antiamyloidogenic properties of this compound.

In this study, human insulin was used as protein model to investigate the antiamyloidogenic properties of HT. In this respect, insulin is a useful and well-studied model protein as it is able to form amyloid aggregates both in vitro and in vivo [22,23,24]. Amyloid deposits involving insulin fibrils have been observed both in patients with type II diabetes, in normal aging and after repeated insulin injections [22,25,26,27]. Specifically, insulin-derived amyloidosis (IDA) is a pathological complication of insulin therapy in which amyloid deposits are formed in the insulin injection site triggering an inflammatory response and necrosis in the surrounding tissue [27,28,29]. Moreover, IDA causes poor glycemic control and increased insulin dose requirements because of the impairment in insulin absorption [30].

Insulin amyloid formation has been characterized in different conditions in vitro and it is known to be affected by several factors, such as pH, temperature, protein concentration, ionic strength, presence of denaturants and agitation [31,32,33,34,35,36]. As well as other amyloidogenic proteins, insulin amyloid aggregation proceeds by stepwise oligomerization, nucleation and growth phase and its amyloid cytotoxicity is mainly associated to the oligomeric species while amyloid fibrils are almost harmless [34,37,38,39]. Insulin is a small protein composed of 21 residues in the A-chain and 30 residues in the B-chain connected by two disulfide bridges and it is known to exist in solution as an equilibrium mixture of different association states strongly affected by the environmental conditions [23,40,41,42]. This protein is mainly organized in an α-helical structure and its amyloid aggregation is proposed to occur via partial unfolding of a monomeric intermediate that promotes protein oligomerization and the α to β transition underlying the amyloid fibril formation [43,44,45]. Considering the importance of physiological conditions, in our study, insulin amyloid formation was performed under conditions more closely resembling physiological pH and temperature (pH 7.0, 37 °C). Our results show that HT is able to fully inhibit insulin amyloid formation and, consequently, the production of amyloid toxic species. Interestingly, the antiamyloidogenic effect of HT seems to be ascribed to the stabilization of the insulin monomeric state.

## 2. Results

### 2.1. Inhibitory Effect of HT on Insulin Amyloid Aggregation

Insulin represents a widely used model protein in the study of amyloid formation as it is able to form amyloid aggregates both in vitro and in vivo [22,23,24]. Insulin amyloid formation is affected by several factors, such as pH, temperature, protein concentration, ionic strength, and the presence of denaturants [31,32,33,35]. Considering the importance of physiological pH and temperature, in this study, insulin amyloid aggregation was performed at pH 7.0 under agitation with teflon beads at 37 °C.

To evaluate the effect of HT on insulin amyloid formation, Thioflavin T (ThT) fluorescence, far-UV CD spectroscopy and cytotoxicity studies have been performed in the presence and in the absence of HT and analyzed at different times of incubation in aggregating conditions. The HT concentrations were chosen at four different molar ratio respect to human insulin—0:1, 0.5:1 (0.5×), 1:1 (1×) and 2:1 (2x). Insulin amyloid formation at pH 7.0 under agitation with teflon beads at 37 °C has been well-characterized in our group. In particular, in these conditions, the full conversion from a predominant α-helical structure, that characterizes the native protein, to the cross β-structure of the amyloid fibrils is known to take place in 24 h [35,46].

At first, the effect of HT on insulin amyloid formation was evaluated by ThT binding assay. ThT is an amyloid specific dye as it shows a strong fluorescence emission only when bound to the cross-β amyloid structure [47,48]. Figure 2 reports the amyloid aggregation kinetics of human insulin in the absence and in presence of HT (0.5×, 1×, 2×) evaluated by ThT assay.

As expected, the sample incubated in the absence of HT shows the higher fluorescence intensity already at 24 h incubation, indicating full amyloid fibril formation at this time point. Interestingly, the samples incubated with equimolar amount (1×) or molar excess (2×) of HT do not show ThT fluorescence in time, thus indicating that, at these molar ratios, HT is able to inhibit amyloid fibril formation in human insulin. These samples were followed in time up to two weeks and no fluorescence intensity was detected at all time points. In comparison, the sample incubated with a substoichiometric amount of HT (0.5×) showed a reduced fluorescence intensity at 24 h, thus suggesting a partial inhibition of amyloid fibril formation. The slower aggregation kinetics observed for this sample could be likely due to a reduced concentration of amyloidogenic protein in solution.

In order to monitor the effect of HT on the structural transition underlying amyloid formation in human insulin, far-UV CD spectra of the protein in the absence and in the presence of HT (0.5, 1 and 2x) were recorded at 0 and 24 h of incubation in aggregating conditions (Figure 3).

At the beginning of the aggregation process, the spectrum of human insulin almost resembles that of the native protein, showing two minima at 222 and 208 nm, and a positive signal around 195 nm consistent with the presence of α-helical conformation. Interestingly, the spectra recorded in the presence of HT are fully overlapping with that of insulin, thus suggesting that the presence of HT does not affect secondary structure content in human insulin (Figure 3A). After 24 h of incubation in aggregating conditions, while the CD spectrum of insulin is characterized by a clear minimum at around 218 nm characteristic of extensive β-sheet structures, the sample aggregated in the presence of HT 1 and 2× is still maintaining the native helical structure (Figure 3B). These data indicate that, as expected, at 24 h of incubation in aggregating conditions, insulin undergoes a conformational transition from an α-helix to a β-sheet structure, associated to the amyloid fibril formation. Differently, for the sample incubated in the presence of HT in the molar ratio 1 and 2×, the α to β-transition is inhibited, thus indicating that HT, when in equimolar amount, is able to stabilize the native helical structure and impair the conformational transition underlying amyloid formation. These results are in perfect agreement with the ThT data and suggest that an equimolar amount of HT stabilizes insulin native structure, thus inhibiting the conformational changes underlying amyloid fibril formation.

### 2.2. Effect of HT on Insulin Amyloid Toxicity

Further confirmation of HT inhibition in insulin amyloid formation was obtained by cellular toxicity studies. Indeed, amyloid aggregates are known to affect cell toxicity. In particular, it is well-established that the most toxic amyloid species are the early soluble oligomeric aggregates whereas amyloid fibrils are essentially harmless [37,38,39,49,50]. Furthermore, for insulin, the higher cytotoxicity is associated with the oligomeric species, formed at 12 h in our aggregation conditions and poorly ThT-reactive, while amyloid fibrils (formed at 24 h), which are highly ThT-reactive, are almost harmless [34]. To assess the cytotoxicity of insulin species formed in the presence of HT in aggregating conditions, the cell viability was evaluated in neuroblastoma cells by the MTT assay, which measures the cellular metabolic activity (Figure 4). To this aim, SH-SY5Y cells were exposed for 24 h to insulin incubated for 0, 12 and 24 h in aggregating conditions in the presence of different concentrations of HT (0, 0.5, 1, 2×). As previously observed, in the absence of HT, early amyloid aggregates (12 h) reduce cell viability by approximately 50% compared to untreated cells, while amyloid fibrils (24 h) are not toxic [34,46]. Interestingly, insulin samples aggregated in the presence of HT showed no cell toxicity at any incubation time thus indicating the absence of amyloid oligomeric species in these samples. The sample incubated in the presence of 0.5x HT was not showing cytotoxicity either at 12 and 24 h incubation and this could be due to a low concentration of toxic oligomeric species at these time points, as suggested by the low ThT reactivity. These data clearly show that the presence of an equimolar amount HT does not promote the formation of soluble cytotoxic species in human insulin, thus suggesting that HT could have a protective effect in amyloid toxicity.

### 2.3. Effect of HT on Insulin Oligomerization State

Insulin amyloid aggregation is known to be strongly related to the native protein oligomerization state [44,51,52,53]. Native insulin has been described as an equilibrium mixture of different association states in solution and the ratio between these species is known to be strongly affected by the environmental conditions. Specifically, insulin is mainly monomeric at pH 2.0 in 20% acetic acid, dimeric in the pH 2–8 range in zinc-free solution, and hexameric at pH 7.5 in the presence of zinc [23,40,41,42]. In our experimental conditions (pH 7.0, 37 °C), insulin amyloid is believed to be triggered by the dimeric species highly stabilized in solution at this pH value [45]. In order to probe the effect of HT on the oligomerization state of human insulin in solution, we have firstly analyzed the CD signal in the near-UV region of insulin in the absence and in the presence of different concentrations of HT (0.5, 1, 2×) at pH 7.0 (Figure 5). The CD activity of insulin in this region originates mostly from tyrosyl side chains as this protein lacks tryptophan and contains four tyrosine and three phenylalanine residues. Interestingly, the four tyrosines are distributed over three surfaces of the molecule differently involved in dimer formation and higher order oligomer association. Specifically, TyrB16 and TyrB26 are fully exposed in the insulin monomer and shielded in the dimer, being located at the dimer interface, while the exposure of the other tyrosines (TyrA14, TyrA19) is not affected by dimer formation [54]. For this reason, the CD signal at 275 is strongly affected by the insulin association state. In particular, the signal intensity is enhanced disproportionally as monomers interact to form dimers and as dimers interact to form hexamers, with greater effect associated to the first process. This is due to the fact that insulin dimerization is likely associated with the largest structural changes, whereas the formation of tetramers or higher order oligomers should lead to the smallest structural perturbations [55,56].

In Figure 5, in the absence of HT, the insulin near-UV CD spectrum at pH 7.0 mostly resembles that reported by Uversky and coworkers [56], and it is mainly associated with the insulin dimeric state highly stabilized at this pH value. Interestingly, in the presence of HT, even in equimolar concentration, a reduction in the spectrum intensity occurs, thus suggesting that this molecule could stabilize a different oligomerization state (likely a monomer) in human insulin. Moreover, the near-UV CD spectra were also recorded for insulin incubated in aggregating conditions in the presence of HT (data not shown). Surprisingly, the spectra of insulin incubated with HT (1 and 2×) for 48 h were fully overlapping with those recorded at the beginning of the process (reported in Figure 5), thus suggesting that HT is able to stabilize insulin monomers even in aggregation conditions. These data suggest that HT is likely to shift the equilibrium towards the monomeric insulin species, thus inhibiting protein oligomerization and amyloid aggregation.

To further explore this hypothesis, we evaluated the hydrodynamic size of insulin in the absence and in the presence of HT by dynamic light scattering (DLS). In Figure 6, the second-order autocorrelation functions g_2_(τ) for 2 mg/mL insulin alone and with 2× HT are reported. From the fit to single decaying exponentials, the resulting apparent diffusion coefficients D_app_ are (6.4 ± 0.1) × 10^−7^ cm^2^ s^-1^ for insulin, and (9.8 ± 0.1)x10^-7^ cm^2^ s^-1^ for insulin incubated in the presence of HT, which would correspond to apparent hydrodynamic radii R_h,app_ of (3.84 ± 0.04) nm and (2.51 ± 0.05) nm, respectively. Insulin apparent R_h,app_ in the absence of HT reflects the presence of hexamers, with a literature reported value of 5.6 nm [57], that, at similar concentrations and pH values, have been detected in zinc-free solutions [42]. Although this analysis does not provide quantitative information on the oligomeric size distribution, we can confidently affirm that, in the presence of HT, the equilibrium shifts towards lower size insulin species, which would inhibit for further assembly and aggregation.

### 2.4. HT-Insulin Interaction

In order to unveil the molecular determinants of HT inhibition in insulin amyloid aggregation, the interaction between insulin and HT in native conditions was investigated by fluorescence spectroscopy. Insulin only contains four tyrosyl residues as fluorescence emitters and, for this reason, its spectrum is characterized by the typical tyrosyl emission at 305 nm. To probe the specific interaction between native insulin and HT, we have tested the ability of HT as a tyrosine quencher (Figure 7). Figure 7A shows the emission fluorescence spectra of insulin in the absence and in the presence of HT at different molar ratios. As expected, the insulin emission spectrum shows a strong fluorescence intensity centered at 305 nm. Interestingly, upon addition of increasing concentrations of HT, the fluorescence intensity regularly decreases, thus indicating that this molecule induces fluorescence quenching of tyrosine emission. The fluorescence quenching of a protein induced by small molecules may be collisional or static if the molecule specifically interacts with the protein. In order to exclude a collisional quenching by HT, the experiment was performed both for native insulin and monomeric tyrosyl residues (N-acetyl-L-tyrosine ethyl ester). Figure 7B shows the Stern–Volmer plot of tyrosine quenching by HT obtained both for insulin and free tyrosine. The F_0_/F values recorded for free tyrosine were significantly lower compared to those recorded for insulin at each molar ratio, thus suggesting that no collisional quenching is involved and indicating the formation of a specific insulin-HT complex. These data indicate that HT specifically interacts with human insulin in a region close to the tyrosyl residues but, as no full quenching of tyrosyl fluorescence is observed, we can hypothesize that the HT binding only affects few tyrosines residues in insulin.

## 3. Discussion

The presence of amyloid aggregates is considered a key feature in the pathological signs of many neurodegenerative diseases. As a consequence, much effort has been devoted to the identification of new molecules able to interfere with the aggregation process in amyloidogenic proteins, also preventing the appearance of toxic species (soluble oligomers and protofibrils). Recently, several studies have suggested that natural phenolic compounds can interfere with the amyloid aggregation process both by affecting the amyloid intermediate formation and promoting the aggregation cascade towards nontoxic species [58,59]. In particular, the main phenolic compounds of extra-virgin olive oil, i.e., oleuropein aglycone (OleA) and its main metabolite, hydroxytyrosol (3,4-dihydroxyphenylethanol (HT), aside from their biological and pharmacological properties, have been shown to affect the amyloid aggregation process in different model proteins in vitro [12,15,16,17,18,19,20,21,60]. Because only little information is available for HT compared to OleA, in this study, we tested the molecular effect of HT in the amyloid aggregation process of human insulin.

Insulin is a useful and well-studied model protein whose amyloid-like fibril formation can be induced under experimental conditions that destabilize the native state and promote protein oligomerization underlying amyloid fibril formation [43,44,45]. Our results demonstrate that HT is a potent and promising inhibitor of insulin amyloid aggregation and this property seems to be associated with the stabilization of the insulin native structure. Indeed, both far-UV CD analysis and ThT assay indicate that HT (even in equimolar concentrations) is able to inhibit the α to β transition underlying amyloid assembly and keep the protein in the native state. Also, our data indicate that HT completely reverses the toxic effect produced by amyloid insulin aggregates and this can be explained by the ability of HT to fully inhibit amyloid formation and, consequently, the production of toxic species. The inhibition of insulin aggregation induced by HT is likely mediated by a specific interaction of this molecule with regions very close to tyrosyl residues of the protein, as suggested by the fluorescence quenching studies. 

Insulin amyloid aggregation is known to be strongly related to the protein oligomerization state [44,51,52,53]. Native insulin is known to exist as an equilibrium mixture of different association states in solution and the ratio between these species is strongly affected by the environmental conditions. For instance, insulin is predominantly monomeric at pH 2.0 in 20% acetic acid, dimeric in the pH 2–8 range in zinc-free solution, and hexameric at pH 7.5 in the presence of zinc [23,40,41,42]. Insulin amyloid aggregation is strongly affected by its oligomerization state and, in particular, at pH 7.0 without zinc, it is believed to be triggered by the dimeric species highly stabilized in solution at this pH value [45]. Interestingly, both near-UV CD spectroscopy and DLS analysis indicate that the HT binding to insulin might strongly stabilize the insulin monomeric state in solution and this could explain the inhibition of oligomerization underlying fibril formation. Specifically, the CD signal in the near region is strongly affected by insulin association state as this protein contains four tyrosines residues differently exposed in monomers, dimers and high order oligomers [55,56]. Our CD results suggest that the presence of HT induces a different exposure of TyrB16 and TyrB26 in the insulin structure that is likely associated with the equilibrium shift from a dimeric to monomeric state. Also, the evaluation of the hydrodynamic size of insulin performed by DLS indicates that the presence of HT shifts the equilibrium towards lower size insulin species, like monomers, unable to further assemble. The overall data suggest that HT could specifically interact with human insulin in a region close to Tyr16 or Tyr26 of the B-chain, also providing a rationale for the partial tyrosine fluorescence quenching observed in the presence of HT. As TyrB26 is located in an unstructured region responsible for insulin dimerization (Figure 8A), we could hypothesize that the stabilization of this unstructured region by HT might induce inhibition of protein oligomerization and, consequently, amyloid aggregation in human insulin (Figure 8B). Further studies will be needed to better clarify this mechanism.

Recently, HT has been shown to affect the amyloid aggregation process in hen egg white lysozyme, Aβ_1-42_, Amylin, Tau protein and α-synuclein [15,16,17,18,20,21]. In particular, HT has been suggested to interact with these amyloidogenic proteins and affect amyloid intermediate formation as well as promote the formation of off-pathway species unable to induce cytotoxicity. Interestingly, our results suggest, for insulin, a different effect compared to other model proteins. Indeed, for the first time, HT is shown to induce a complete inhibition of amyloid formation mediated by a structural stabilization of the native structure. In this respect, our study suggests that the molecular effects of HT in amyloid aggregation are strongly dependent on molecular interactions between this compound and specific regions exposed on the protein structure. However, although different effects of HT could be observed in relation to different polypeptide chain properties, so far, HT has always been shown to produce positive effects in amyloid aggregation as promoting the appearance of nontoxic species.

Moreover, considering that HT is known to induce neuroprotective effects both in vitro and in vivo through a combination of antioxidant and anti-inflammatory actions [7,60,61,62,63], our findings suggest that the beneficial effects observed for HT may arise from multifunctional activities. These include, among others, their ability to interfere with the aggregation pathways of disease-associated proteins, their antioxidant/anti-inflammatory power, and their proautophagic effect [7,60,61]. In addition, plant polyphenols are known to induce neuroprotection through the activation of hormetic pathways, including vitagenes network, a complex of redox-dependent genes, which sense redox perturbations, such as oxidative damage and inflammation, and actively operate in promoting cell survival under physiopathological conditions. In this respect, the restoration of redox homeostasis by olive polyphenols suggests a potential therapeutic target in the crosstalk of the inflammatory response process and oxidative stress in neurodegeneration [64,65,66]. Taken together, the molecular determinants of these activities provide a solid, yet still incomplete, rationale supporting the suggested protection against amyloid-associated neurodegeneration by olive polyphenols. In this respect, our data further contribute to make HT a promising compound for novel therapeutic strategies in the prevention and/or treatment of amyloid-associated diseases, especially taking into account its bioavailability and the ability to cross the blood brain barrier [60,67,68].

## 4. Materials and Methods

### 4.1. Materials

Human insulin, hydroxytyrosol, Thioflavin T, N-acetyl-L-tyrosine-ethyl ester, 3-(4,5-dimethylthiazol-2-yl)-2,5-diphenyl-tetrazolium bromide (MTT) (Sigma-Aldrich Co., St. Louis, MO, USA). All other chemicals were of analytical grade. 

### 4.2. Insulin Preparation and Amyloid Aggregation

Human insulin 4 mg/mL was dissolved in ultrapure milliQ water at pH 2.0 and the concentration was estimated by absorbance (ε_275_= 4560 M^−1^cm^−1^). Insulin was then diluted at 2 mg/mL concentration in a phosphate buffer of 50 mM, pH 7.0 [69]. HT was dissolved in ultrapure milliQ water at 100 mM concentration. For aggregation studies, protein samples at a final concentration of 1 mg/mL were incubated at 37 °C under vigorous agitation with teflon beads, 1/8″ diameter (Polysciences, Inc., Warrington, PA, USA) in the absence and in the presence of HT in molar ratios to insulin of 0.5, 1, and 2x. Aliquots of protein were collected in sterile conditions and immediately analyzed.

### 4.3. Circular Dichroism Measurements

CD spectra were recorded at 25 °C on a JascoJ-715 spectropolarimeter using thermostated quartz cells of 0.1 cm. The spectral acquisition was taken at 0.2 nm intervals with a 4 sec integration time and a bandwidth of 1.0 nm. Each spectrum represents an average of five scans and photomultiplier absorbance never exceed 600 V in the analyzed region. All measurements were performed under nitrogen flow and the working protein concentration was 0.3 mg/mL for far-UV measurements and 2 mg/mL for near-UV. Data were corrected for buffer contributions using the software provided by the manufacturer (System Software version 1.0, Easton, MD, USA) and transformed in mean residue ellipticity.

### 4.4. Fluorescence Measurements

ThT fluorescence emission was monitored in time using a 96-well Thermo Scientific Fluoroskan Ascent F2Microplate with thermostatic control at 37 °C. ThT working concentration was 20 μM for all samples, and with excitation and emission wavelengths of 450 and 485 nm, respectively. All measurements were performed in triplicate. The time course of fluorescence emission of ThT incubated with only HT at the different concentrations used for experiments with insulin was subtracted from the correspondent kinetic profile. Tyrosine fluorescence experiments were performed on a Perkin Elmer Life Sciences LS 55 (Boston, MA, USA) spectrofluorimeter and tyrosine emission (λ_ex_ 275 nm/λ_em_ 305 nm) was evaluated in both insulin and free tyrosine after addition of HT at different insulin:HT molar ratio (1:0, 1:0.5, 1:1, 1:2, 1:3). Tyrosyl fluorescence quenching was monitored by estimation of the F_0_/F ratio considering the fluorescence intensity at 305 nm of the sample before (F_0_) and after (F) the addition of HT. Working concentrations were 10 µM for insulin and 40 µM for free tyrosine.

### 4.5. Dynamic Light Scattering

The samples, freshly prepared and filtered using 0.2 µm Millex-GV filters, were placed into a dust-free quartz cell and kept at 20 °C in the thermostated cell compartment of a Brookhaven Instruments BI200-SM goniometer. The temperature was controlled within 0.1 °C using a thermostated recirculating bath. The light scattered intensity and time autocorrelation function were measured at θ = 90° by using a Brookhaven BI-9000 correlator and a 50 mW He–Ne laser tuned at λ = 632.8 nm. At each condition, the electric field intensity autocorrelation function g_2_(τ) data could be interpolated within experimental precision by a single exponential expression as
*g*_2_(*τ*) = *α* + *β**exp* (−2*D_app_**q*^2^*τ*)(1)
where D_app_ represents the z-average translational diffusion coefficient and q is the scattering vector $q=\frac{4\pi n_{0}}{\lambda_{0}}\left(\frac{\theta }{2}\right)\;$ with n_0_ the solvent refractive index and λ_0_ the incident wavelength [42]. The apparent hydrodynamic radius *R*_h,app_ can be obtained from the translational diffusion coefficient (*D_app_*) using the Stokes–Einstein relationship $R_{h,app}=\frac{k_{B}T}{6\pi \eta D_{app}}$ where k_B_ is the Boltzmann constant, T the absolute temperature and η the viscosity. To assess reproducibility, each experiment was repeated at least three times using independent batches of proteins and data are averaged over the measurements.

### 4.6. Cell Cultures and Treatments

SH-SY5Y human neuroblastoma cells (ATCC# CRL-2266) were cultured in Eagle’s Minimum Essential Medium (EMEM) supplemented with 10% fetal bovine serum, 3.0 mM glutamine, 50 units/mL penicillin and 50 mg/mL streptomycin in a 5.0% CO_2_ humidified environment at 37 °C. Cells were exposed for 24 h to 30 µM insulin incubated for 0, 12 and 48 h in aggregating conditions in the presence of HT at different concentrations (0, 1, 2×). Cells in culture medium without protein and in the presence of HT at the tested concentrations served as a control.

### 4.7. MTT Assay

Cell viability was evaluated by the inhibition of the ability of cells to reduce the metabolic dye 3-(4,5-dimethylthiazol-2-yl)-2,5-diphenyltetrazolium bromide (MTT) to a blue formazan product, as previously reported [70]. After treatments, cells were rinsed with phosphate buffer solution (PBS) and incubated with 0.5 mg/mL MTT in cell medium for 3 h at 37 °C. Upon removal of the medium, cells were treated with isopropyl alcohol, 0.1 M HCl for 20 min and samples and the MTT reduction was estimated by measuring the difference in absorbance between 570 and 690 nm. Data are expressed, as average percentage reduction of MTT with respect to the control ±S.D. Data represent an average from five independent experiments carried out in triplicate.

### 4.8. Statistical Analysis

Statistical analyses were performed using Stata software (Version 13.0; StataCorp LP., College Station, TX, USA). Tukey’s post hoc test was used if the treatment was significant on analysis of variance (ANOVA). All data are represented as the mean ± SE. Statistical significance was set at *p* < 0.05.

### 4.9. Data Availability

The datasets generated and/or analyzed during the current study are available from the corresponding author on reasonable request.

## Figures and Tables

**Figure 1 ijms-21-04636-f001:**
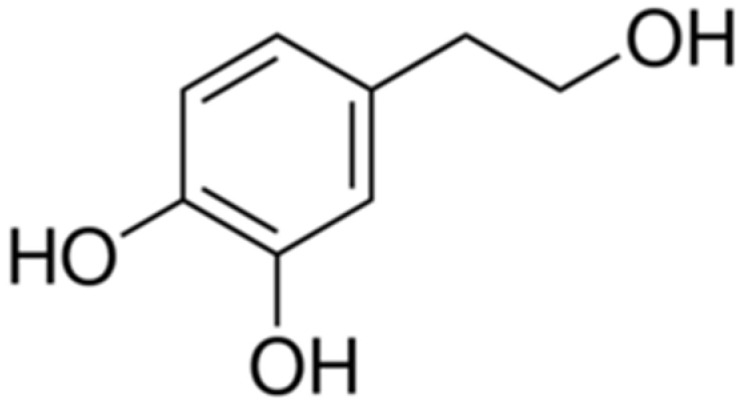
Chemical structure of hydroxytyrosol (HT).

**Figure 2 ijms-21-04636-f002:**
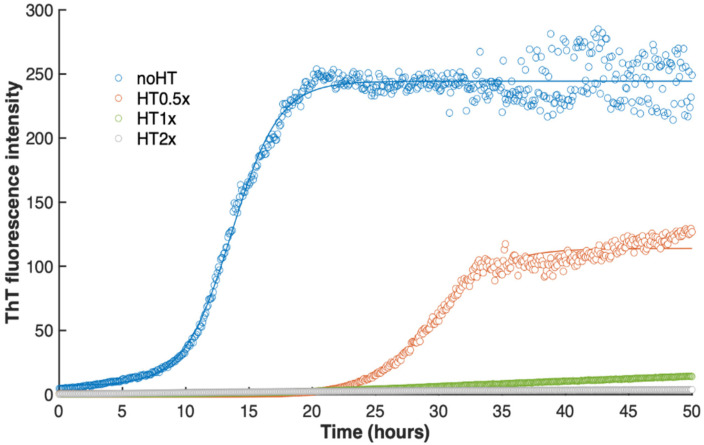
Effect of HT on insulin amyloid aggregation by Thioflavin T (ThT) assay. ThT fluorescence intensity recorded for insulin in aggregating conditions in the absence and in the presence of HT at 0.5, 1, and 2 x molar ratio. ThT emission was recorded at 482 nm upon excitation at 450 nm. The results are averages over triplicates for each sample condition. The solid line, reported as a guide for the eye, is the best fit to the data given by the sigmoidal equation f = y_0_ + a/(1 + exp(x_0_ − x)^c). Other experimental details are described in the Methods section.

**Figure 3 ijms-21-04636-f003:**
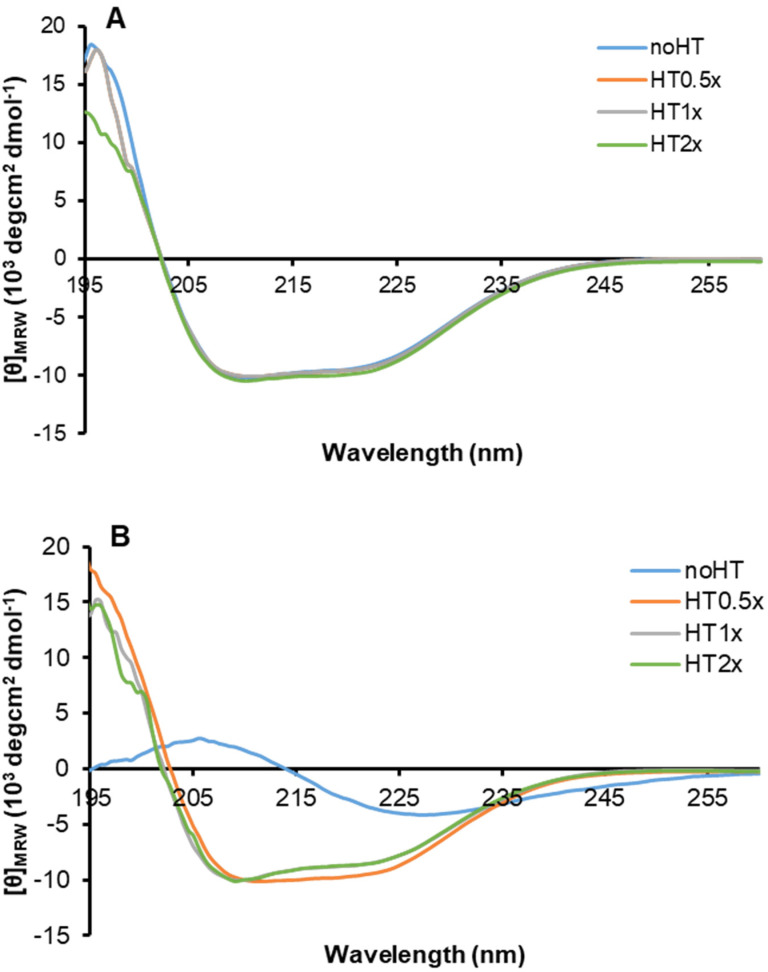
Effect of HT on insulin amyloid formation. Far-UV CD spectra of insulin in aggregating conditions in the absence and in the presence of HT 0.5, 1, and 2× molar ratio at 0 (**A**) and 24 (**B**) hours of incubation. Protein concentration was 0.3 mg/mL, other experimental details are described in the Methods section.

**Figure 4 ijms-21-04636-f004:**
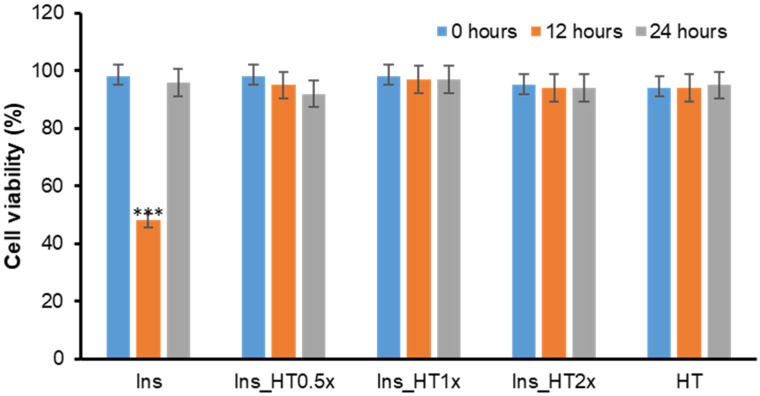
Effect of HT on insulin amyloid toxicity. Cytotoxicity of insulin amyloid species formed in the absence and in the presence of HT. SH-SY5Y cells were exposed for 24 h to insulin incubated for 0, 12 and 48 h in aggregating conditions in the absence and in the presence of HT (0.5, 1, and 2x) and cell viability was evaluated by MTT assay. Data are expressed as average percentage of MTT reduction ±SD relative to control cells from triplicate wells from 5 separate experiments (*p* < 0.01). ***: *p* < 0.001. HT represents cells exposed to HT at the higher working concentration. Other experimental details are described in the Methods section.

**Figure 5 ijms-21-04636-f005:**
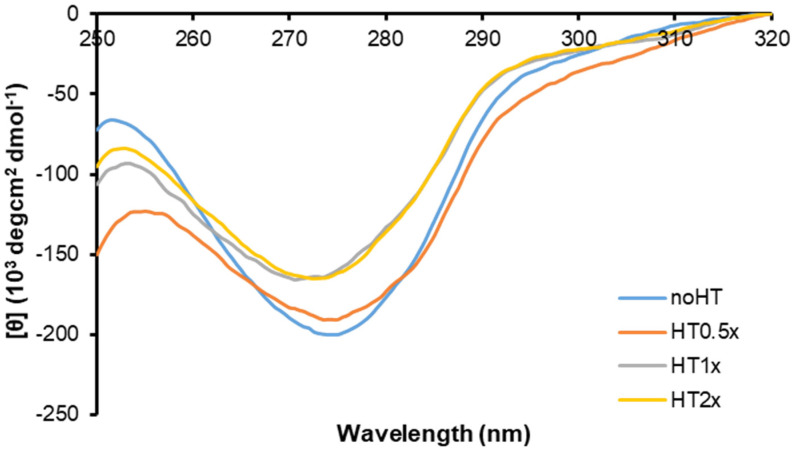
Effect of HT on insulin oligomerization state monitored by near-UV CD spectroscopy. Near-UV CD spectra of insulin in the absence and in the presence of HT in a 0.5, 1, and 2× molar ratio. Protein concentration was 2.0 mg/mL, other experimental details are described in the Methods section.

**Figure 6 ijms-21-04636-f006:**
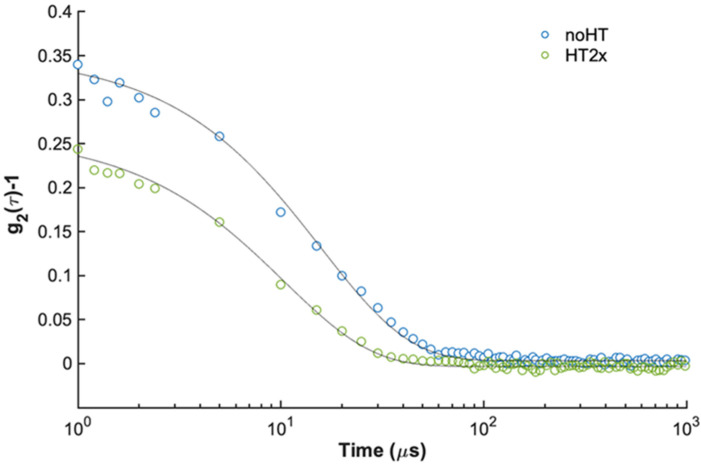
Effect of HT on insulin oligomerization state monitored by dynamic light scattering (DLS). Second order autocorrelation functions g_2_ (τ) measured for 2 mg/mL, pH 7.0 insulin incubated in the absence (blue) and in the presence (green) of HT in a 2× molar ratio. Continuous lines are the best fit to single exponential functions as described in Equation (1).

**Figure 7 ijms-21-04636-f007:**
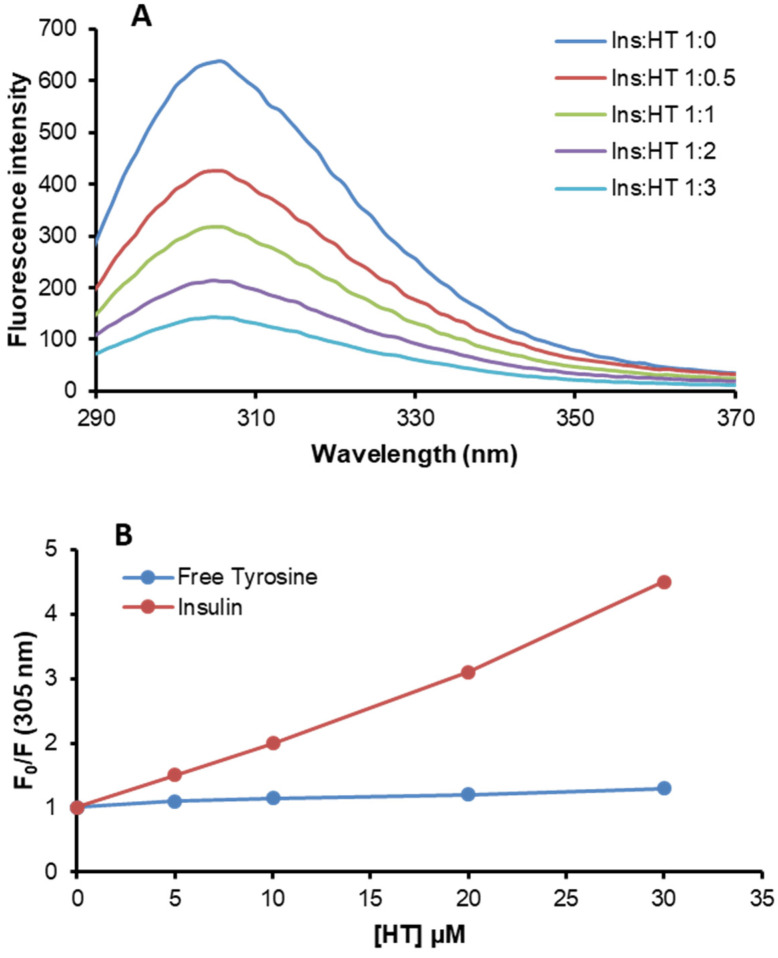
Insulin-HT interaction monitored by fluorescence spectroscopy. (**A**) Tyrosine fluorescence emission was evaluated in both insulin and free tyrosine after addition of HT at different molar ratios (1:0, 1:0.5, 1:1, 1:2, 1:3). (**B**) Stern–Volmer plot of tyrosine quenching by HT. Working concentrations were 10 µM for insulin and 40 µM for free tyrosine. Other experimental details are described in the Methods section.

**Figure 8 ijms-21-04636-f008:**
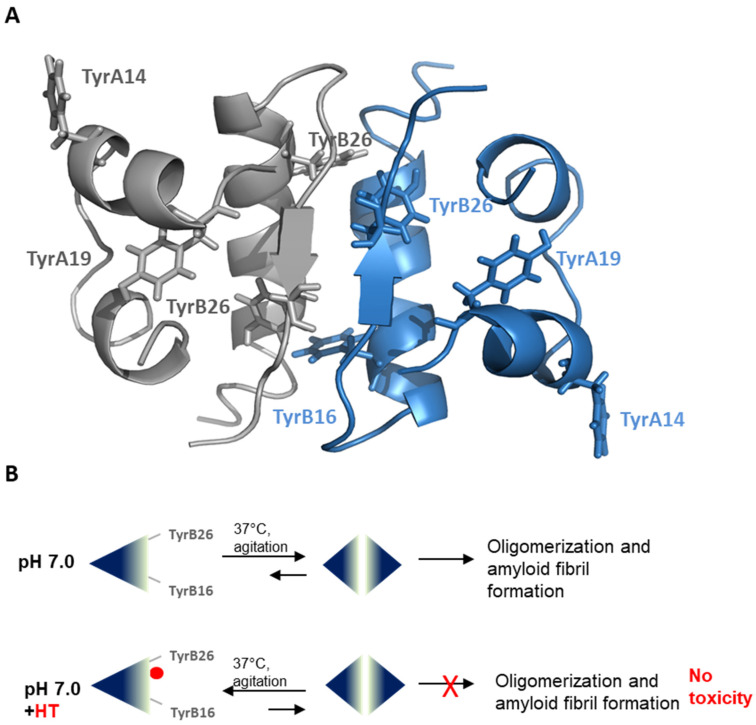
Working hypothesis for HT inhibition in insulin amyloid formation. (**A**) Structural representation of human insulin in its dimeric form (PDB code 3AIY). The two monomers are represented in gray and light blue and tyrosine residues are represented in the tridimensional structure for each monomer. (**B**) Schematic representation of the effect of HT on insulin amyloid formation.

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
