# Peer review of "Hydroxytyrosol Inhibits Protein Oligomerization and Amyloid Aggregation in Human Insulin"

_ijms, 2020, doi:10.3390/ijms21134636_

Round 1

Reviewer 1 Report

This is an interesting paper. The study is well-conceived and well-performed. This reviewer is satisfied with the relevance of this work and its potential implications for human health. Olive oil polyphenols, in particular hydroxytyrosol is known to posses various beneficial properties in different diseases but it is interesting to note that hydroxytyrosol is able to inhibit toxicity caused by protein amyloid aggregation in vitro. However, although the content presented are convincing, the work raises some concerns which will need to be addressed. The questions posed are of extremely high interest, and the paper gives adequate definitive information, therefore pending addressing one minor question is possible to accept for publication.

Minor concerns:

Please revise the references in lines 404 pag 12, 407 pag 12 and 475 pag 14, insert the name of all authors.

Add at line 298 pag 8 these recent papers to explain better the neuroprotective effects of olive oil polyphenols in vivo during neurodegeneration: Brunetti et al., Int J Mol Sci. 2020 8: 21-2588. Di Rosa et al., Int J Mol Sci. 2020.May 29;21(11):E3893. 

Author Response

We are grateful to the Reviewer for the suggestions. The manuscript has been modified accordingly.

Reviewer 2 Report

This is a highly interesting paper. The study is well-conceived and well-executed. This reviewer is satisfied with the significance of this study, the care in which the study was performed, and the implications of the results for human health. However, although the results presented are convincing, the work raises some concerns which will need to be addressed. The questions posed are of extremely high interest, and the paper gives adequate definitive information, therefore pending addressing one minor question is possible to accept for publication.

Minor concerns:

1. Given the relationship between oxidant/antioxidant status and the vitagene network and its possible biological relevance in neuroprotection, Authors while interpetrating results should discuss appropriately this aspect and make proper connection with emerging principles of hormesis ( Calabrese et al., Ageing Res Rev. 2018, Mar;42:40-55); Leri et al., Int J Mol Sci. 2020, 21(4):1250; 178. Pilipenko V. et al., J Neurosci Res. 2019 97:708-726; 175; Peters V, et al., Int J Mol Sci. 2018 13;19(9) pii: E2751. doi: 10.3390/ijms19092751.

Author Response

(The authors gave the same response as above.)
